# Leishmaniasis in the United States: Emerging Issues in a Region of Low Endemicity

**DOI:** 10.3390/microorganisms9030578

**Published:** 2021-03-11

**Authors:** John M. Curtin, Naomi E. Aronson

**Affiliations:** 1Infectious Diseases Service, Walter Reed National Military Medical Center, Bethesda, MD 20814, USA; 2Infectious Diseases Division, Uniformed Services University, Bethesda, MD 20814, USA; naomi.aronson@usuhs.edu

**Keywords:** asymptomatic visceral leishmaniasis, autochthonous leishmaniasis, travel-related leishmaniasis, climate change, transfusion transmission, immunosuppression

## Abstract

Leishmaniasis, a chronic and persistent intracellular protozoal infection caused by many different species within the genus *Leishmania*, is an unfamiliar disease to most North American providers. Clinical presentations may include asymptomatic and symptomatic visceral leishmaniasis (so-called Kala-azar), as well as cutaneous or mucosal disease. Although cutaneous leishmaniasis (caused by *Leishmania mexicana* in the United States) is endemic in some southwest states, other causes for concern include reactivation of imported visceral leishmaniasis remotely in time from the initial infection, and the possible long-term complications of chronic inflammation from asymptomatic infection. Climate change, the identification of competent vectors and reservoirs, a highly mobile populace, significant population groups with proven exposure history, HIV, and widespread use of immunosuppressive medications and organ transplant all create the potential for increased frequency of leishmaniasis in the U.S. Together, these factors could contribute to leishmaniasis emerging as a health threat in the U.S., including the possibility of sustained autochthonous spread of newly introduced visceral disease. We summarize recent data examining the epidemiology and major risk factors for acquisition of cutaneous and visceral leishmaniasis, with a special focus on implications for the United States, as well as discuss key emerging issues affecting the management of visceral leishmaniasis.

## 1. Introduction

Leishmaniasis is more common in the United States (U.S.) than most Americans realize, involving both autocthonous and imported infections. The recent deployment of millions of Americans to Iraq and Afghanistan has been associated with thousands of cases of cutaneous leishmaniasis (CL) [1], but even more concerning is the likelihood of large numbers of persons with unrecognized asymptomatic visceral leishmaniasis (VL) [2]. In this review, we will summarize the known epidemiology of leishmaniasis in the U.S., and then focus on a few emerging issues. These include climate change and potential impact on leishmaniasis in the U.S., some emerging epidemiology, diagnostic issues and treatment options; we then provide detailed discussion of issues around asymptomatic visceral leishmaniasis, interaction with immunosuppression, possible long-term consequences, and secondary transmission risks in the U.S.

## 2. Human Leishmaniasis: Epidemiology in the United States

### 2.1. Autochthonous Leishmaniasis

While most leishmaniasis in the U.S. is likely acquired during travel abroad, cutaneous *Leishmania mexicana* is endemic to the United States, reported from Texas and southeast Oklahoma among persons with no travel history in the prior 5–10 years [3,4,5,6,7,8]. Texas is the only state to require public health reporting of leishmaniasis (since 2007) and the median number of cases reported from 2008–2017 is six per year, ranging from 0–13 (including both travel and autochthonous cases). A cross-sectional study of histopathologically confirmed cases in Texas found that only 20% were reported to the Texas Department of State Health Services between 2007–2017; 41 (59%) leishmaniasis cases occurred in persons with no travel outside of the U.S. [7]. However, autochthonous cutaneous leishmaniasis in Texas is not new, with the first case of diffuse cutaneous leishmaniasis (DCL) recorded in 1903, and Kipp reports a total of 80 autochthonous cases by 2020 [9]. Most autochthonous cases occur on exposed areas of the face, head, neck, or upper extremities, and affect both men and women of virtually any age (reported range from 2–89 years) [3,5,6,7]. Lesions are typically few in number, appearing as chronic, painless ulcers or papules usually about 1–2 cm size. *L. mexicana* lesions tend to heal without intervention in months, and in one study 88% re-epithelialized over a median of 14 weeks [10]. Interestingly, two Texas cases confirmed as *L. mexicana* were associated with mucosal leishmaniasis, which is unusual for this species [3]. Occasional presentations consistent with diffuse cutaneous leishmaniasis (DCL; large nodules up to 4.5 cm size in many locations) have been reported, however this is a rare complication associated with *L. mexicana*. Two Texas DCL-like cases were unusual in that the histopathology and/or cellular immune responses were fairly normal [3,9]. Intriguingly, two horses in Florida with no travel history were reported to develop cutaneous leishmaniasis caused by *Leishmania (Mundinia)* species, a form that has been associated with human disease in other countries [11,12].

Visceral leishmaniasis is not considered endemic among humans in the U.S., although canine visceral leishmaniasis (caused by *Leishmania infantum*) is, with a 2% seroprevalence rate among U.S. dogs in 2003. This value is much higher (>20%) among hunt clubs and kennels [13,14,15,16,17]. Sand fly transmission to humans is possible but not confirmed, however vector transmission from infected foxhounds to laboratory hamsters has been reported [18,19] and the widely distributed U.S. sand fly *Lu. shannoni* is a competent vector [20]. A potential concern is the recent identification that 19.5% of Iraq-deployed American soldiers have blood testing which suggests that they have asymptomatic visceral leishmaniasis, including 1% with parasitemia as measured by polymerase chain reaction, persisting up to a decade after return to the U.S. [2]. With over 2.7 million persons having deployed to Iraq and Afghanistan in the past two decades, combined with general immigration from global VL endemic areas to the U.S., there could exist reservoirs for further transmission within North America. This could be the explanation for the North Dakotan two-year-old child with no travel history and no pets, who developed cutaneous (eyelid) *L. donovani-infantum* infection. North Dakota has no known sand fly vectors, however the mother immigrated from endemic Nepal the year prior to his birth and may have been asymptomatically infected, passing on the infection during childbirth via bleeding and trauma/abrasions to the external eyelids [21]. Potential routes of transmission within the U.S. can therefore be via sand fly vectors, vertically (maternal-fetal), inoculation, shared equipment during intravenous drug use, and blood transfusions/tissue transplantation.

### 2.2. U.S. Leishmaniasis Vectors and Reservoirs

#### 2.2.1. U.S. Sand Flies That Can Potentially Transmit Leishmaniasis

The sand fly vector(s) of human leishmaniasis in the U.S. is not definitively proven (Table 1). *Lu. anthophora* is a nest associate of the woodrats that are considered likely reservoirs within Texas, and *L. mexicana* has been successfully transmitted to hamsters via *Lu. anthophora* bites [22]. In Texas, *Lu. anthophora* collected at surveillance sites related to human cases was found positive for *L. mexicana* [9,23]. Generally, *Lu. anthophora* has not been considered anthropophilic and was previously discounted as the vector for human infection, but a recent report challenges this concept after molecular analysis of two *Lu. anthophora* collected on a patient’s property found that they had taken a human blood meal; additionally, 4/190 *Lu. anthophora* collected were *L. mexicana* PCR positive [9]. The more anthropophagic vector species *Lu. diabolica* is commonly found in Texas, and *Lu. shannoni* is present though less frequently; both vectors have been found in the laboratory to be capable of transmitting *L. mexicana* and both have been collected at a case’s residence, though they were not found to be infected with *L. mexicana* in those instances [9,23,24,25,26]. While the avid human biter *Lu. diabolica* is considered the most likely vector for human *L. mexicana* infection in Texas [27], wild-caught leishmaniasis-infected sand flies have not been identified. Regarding visceral *L. infantum* infection, notable experiments allowed *Lu. shannoni* to feed on symptomatic VL-infected dogs, which then transmitted *L. infantum* to uninfected dogs at a low rate (9%) [20]. This suggests that the U.S. has a potential, widely distributed sand fly vector for VL within its borders.

#### 2.2.2. Common Animal Reservoirs for Leishmaniasis in the U.S.

The *Neotoma* species woodrat (colloquially known as gopher rat or pack rat) is a vertebrate reservoir host for *Leishmania mexicana*, and implicated species include *Neotoma micropus* (mainly) and *N. floridiana* in Texas, as well as *N. albigula* in southern Arizona [29,30,31,32,33]. They may develop clinical illness with lesions on the ears and swollen feet when infected with *L. mexicana* [30]. In surveillance studies *L. mexicana* prevalence was high; 14.7% in *N. micropus* in Bexar county, Texas [32], 5.6–27% in four southern Texas locations [31], and 33% among *N. albigula* captured in the greater Tucson, AZ area [33]. In a marker-release-capture study, *L. mexicana* infection persisted in *N. micropus* for an average of 190 days (range 104–379 days) [32] with a mean follow-up of 8.2 months. There is some debate about additional vertebrate hosts; McHugh surveyed 100 cotton rats (*Sigmodon hispidus*), 60 opossums, and 20 armadillos in Texas and found no evidence of leishmaniasis [34]. In northeast Mexico, a systematic ecology survey collected 79 specimens of nine vertebrate species, where the most common samples were *Peromyscus maniculatus* (38%), *Sigmodon hispidus* (30%), and *P. leucopus* (16%). Sixteen *N. micropus* were studied and none were positive for leishmaniasis; 21% of *P. leucopus*, 12.5% of *S. hispidus* and 6% of *P. maniculatus* were polymerase chain reaction (PCR) positive for *L. mexicana* [35]. Additionally, the blood of 267 wild canids in Pennsylvania and Tennessee was surveyed for *Leishmania* antibodies with the recombinant K39 (rK39) immunochromatographic test and 5/267(1.9%) were positive (one red fox, four coyotes) [36].

### 2.3. Potential Impact of Climate Change on Autochthonous Leishmaniasis in the U.S.

Vector-borne diseases are sensitive to climatic conditions such as temperature, humidity and precipitation. Temperature increases greater than 1.5 °C above pre-industrial levels are projected by 2030–2052 [37], and *Leishmania* vector sand flies require increased temperatures for development and survival (20–26 °C, though this seems to vary somewhat by species) [38,39,40,41]. Thus, they may not tolerate freezing over winter. Climate change may impact the geographic distribution of sand fly populations, facilitate migration, affect the length of time the vector is seasonally active, increase insect reproduction rate and the size of the overall insect population, and speed of *Leishmania* parasite development in the sand fly. Sand fly dispersal remains limited by geographic barriers such as mountains, high winds, bodies of water, and the maximal flight distance of the sand fly itself (estimated at approximately 1000 m) [42]. Ecological niche modeling in North America, using *Lu. anthophora* and *Lu diabolica* as vectors and *N. albigula, N. floridiana,* and *N. micropus* as reservoirs, included risk components of predicted available habitat, dispersal ability, and the number of people at risk for potential leishmaniasis exposure. Findings showed a high risk of leishmaniasis spread north of Texas because of climate change, with range shifts of *Lu. diabolica* (reaching the southeast boundary of Canada), *N. floridiana* to the east, and *Lu. anthophora* and *N. micropus* to the west [43]. This analysis predicted that the human exposure to leishmaniasis in the United States will double by 2080 [43,44]. Another North American niche model for climate change scenarios focused on the year 2050 predicted that temperate sand fly species such as *Lu. shannoni* and *Lu. cruciata* will increase with niche shifts to the northwest and northeast and become the vector species providing the most contact risk for the U.S. population [45].

### 2.4. Leishmaniasis in American Travelers

#### 2.4.1. Cutaneous and Mucosal Leishmaniasis

Since leishmaniasis is not a notifiable disease in the U.S. (except for Texas), it is difficult to assess the magnitude of travel-acquired infection. Historically, leishmaniasis has mainly been associated with military deployment, with the largest number of American cases being recorded between 2002–2016 (2040 incident diagnoses of which only 25 were VL, the remainder representing cutaneous infection). Exposure was primarily related to Iraq, and less so Afghanistan, travel [1]. The U.S. military does require reporting of all cases and has a centralized leishmaniasis diagnostic laboratory at the Walter Reed Army Institute of Research. New World cutaneous leishmaniasis due to *Viannia* subspecies has classically been treated with pentavalent antimonial drugs (specifically sodium stibogluconate, Pentostam^®^) under investigational new drug protocols, with just two sources in the U.S. (CDC Drug Services and the U.S. Army Medical Materiel Development Activity). This persisted up until 2007–2010, when off-label liposomal amphotericin for treatment of CL became more accepted in practice [46,47]. The number of patients treated with Pentostam^®^ released from the CDC ranged from 22–41 per year from 2004–2009 (per Dr. Rebecca Chancey). In the American military prior to 2003, about 5–20 patients/year were treated with Pentostam^®^ during the previous decade, usually related to jungle training in Latin America. Since the 2014 approval of oral miltefosine for the treatment of cutaneous and mucosal leishmaniasis due to various *L. Viannia* species., the distributor Profounda estimates that about 30 courses per year are provided in the U.S. for treatment of human leishmaniasis (per Todd MacLaughlan).

There are few systematic accountings of leishmaniasis in American civilian travelers (Figure 1). Some details of the 69 U.S. travelers who acquired CL in Mexico and Central or South America between 1985–1990 were published [48]. Demographic characteristics included: 68% male, age range of 3–64 years, and likely acquisition from one of 14 countries, with 33 (56%) being tied to Mexico or Central America. Median days of travel outside the U.S. were 54 (range 4 days–3 years). Some clusters of cases were described related to forested areas near Puerto Maldonado or Manu National Park in Peru, as well as Tikal National Park in Guatemala. 27 (46%) of persons with CL were conducting field studies or school projects and 19 (11%) studied local birds. Clinically, the mean number of skin lesions/person was 1.4 (range 1–8) distributed primarily on the head/neck and extremities (20% on head/neck, 38% upper extremities, and 34% lower extremities) [48]. A more recent review of 955 international travelers with cutaneous (916 cases) and/or mucosal leishmaniasis (ML, 40 cases) seen between 1997–2017 at a GeoSentinel Surveillance Network clinic (~15 U.S. sites contributed 161 (17%) leishmaniasis cases per Dr. Andrea Boggild) reported similar demographic characteristics. The majority of cases were male (62%), with a median age of 30 (range 1–95 years) and median trip duration 47 day (range 1–12,541 days). Interestingly, 10% acquired leishmaniasis on trips lasting ≤2 weeks (mainly tourists and businessmen). The reason for travel in the GeoSentinel study was: 501 (52.5%) tourists, 119 (12.5%) visiting friends/relatives, 10% immigrants/refugees, and 88 (9%) were researchers/volunteer aid workers and missionaries [49]. The most common destinations associated with a subsequent leishmaniasis diagnosis were Bolivia 156 (18%), Costa Rica 97 (11%), Afghanistan 60 (7%), Spain 54 (6%), and Peru 49 (6%); for mucosal leishmaniasis (ML), travel to Bolivia and Peru was associated with most ML cases and mainly seen in tourists as demonstrated in 29 cases (72.5%) [49]. A recent case series and review of mucosal leishmaniasis in the U.S. involved eight civilians and three military personnel; it is most certainly an underestimate, but an important reminder that ML can occur in travelers even remotely following the presumed period of exposure [50].

#### 2.4.2. Visceral Leishmaniasis

Travel-related visceral leishmaniasis (VL) has been reported in the U.S. military whenever deployments occur to endemic regions. Up to 75 U.S. soldiers developed VL during World War II after travel to the Persian Gulf Command, Sicily and Southern Italy, the French Riviera, the India-Burma theater, and northern Africa [51]. During the Persian Gulf War (1990–1992) Saudi Arabia deployment, eight servicemembers developed oligosymptomatic (so-called viscerotropic) leishmaniasis identified as due to *Leishmania tropica* [52], and in the more recent conflicts in Iraq and Afghanistan, 25 additional servicemembers have been diagnosed with active VL due to *Leishmania infantum* [1,53]. Additionally, 39 of 200 (19.5%) tested soldiers were found to have asymptomatic VL after deployment to Iraq, a diagnosis made using a combination of serology, blood PCR, and/or interferon gamma release assay (IGRA), suggesting that many thousands of returned soldiers may be persistently infected [2].

## 3. Emerging Issues Relevant to Leishmaniasis in the United States

Worldwide, cutaneous leishmaniasis is increasing with a global prevalence of 4,166,621 cases in 2017, including a significant female predominance (2.35 million cases) [54]. The top five countries in the 2017 Global Burden of Disease study with the highest age-stabilized growth in CL cases were Guatemala, Syria, Cameroon, Iraq, and Tajikistan [54]. There has also been a new subgenus added, *Leishmania (Mundinia)*. The recognition of what was previously called *L. enriettii* complex (including *L. siamensis* and *L. martiniquensis*) extends areas of travel risk for both cutaneous and visceral leishmaniasis, with *L. (Mundinia) orientalis* in Thailand and Myanmar [55] and *L. (Mundinia) martiniquensis* in the Caribbean joining *L. amazonensis* and *L. waltoni* as a cause of autocthonous CL in the Dominican Republic, Guadelupe, Martinique, Grenada, and Trinidad and Tobago [56]. On a cautionary note, the molecularly different *Leishmania* look-alike *Crithidia* parasites were recently implicated in a fatal illness similar to VL in an immunocompetent 64-year-old man in Brazil, which was unresponsive to three courses of liposomal amphotericin [57].

There are a few diagnostic advances to summarize. While expensive at $2000 per test, thus restricting use to visceral leishmaniasis, the commercially available Karius^®^ test detects cell free DNA (cfDNA) using metagenomic sequencing and a computational data analysis in a plasma sample [58]. This test is validated for detecting *L. infantum and L. donovani* with a limit of detection of 33–74 molecules of microbe specific cfDNA (95% at 41 molecules per microliter), but is not yet FDA-approved/cleared. In the U.S., parasite identification has become increasingly molecular-based rather than determined by classical cellular acetate electrophoresis. A new approach is matrix-assisted laser desorption/ionization (MALDI) mass spectroscopy, which is now being used in clinical laboratories [59].

Treatment options for cutaneous leishmaniasis in the U.S. remain limited, a few recent changes are highlighted. The CDC has revised their protocol for investigational new drug use of sodium stibogluconate to allow intralesional as well as systemic use. However, GlaxoSmithKline is no longer producing Pentostam^®^, thus requiring identification of a new source, a process that is currently pending. The U.S. military suspended their development of topical paromomycin cream, and expanded access is not currently available. Photodynamic therapy and thermotherapy continue to look promising for treatment of Old World cutaneous leishmaniasis [60,61]. On the horizon, the Drugs for Neglected Diseases Initiative is exploring a nitroimidazole 0690 for VL and a modulatory CpG oligonucleotide (CpG-D35) for complicated CL to improve efficacy of parasite-directed chemotherapy [62,63]. The recognized role of trained immunity in human leishmaniasis (mediated by IL-32γ) invites consideration of the nutritional supplement, B-glucan, as an intervention to boost host defenses, although on review no clinical trials were identified [64,65].

## 4. Asymptomatic Visceral Leishmaniasis: Emerging Issues in the United States

### 4.1. Introduction

Visceral leishmaniasis is a spectrum of chronic infection from asymptomatic (latent) to oligosymptomatic (e.g., viscerotropic leishmaniasis) to symptomatic disease, likely resulting from an interplay between the host’s immune response to contain the parasite and the amount of *Leishmania* protozoa in the blood/tissues. In this respect, an analogous comparison can be drawn to tuberculosis, another disease wherein many persons may be infected, however most will only manifest with positive skin tests and while far fewer will have active disease [52,66,67,68]. *Leishmania* infection is chronic, likely persisting for the life of the host. Most VL infections remain subclinical, with overt symptomatic disease seen mainly in infants, young children, and immunocompromised hosts. Symptomatic leishmaniasis is generally associated with levels of up to 200,000 parasites/mL of blood, while asymptomatic leishmaniasis may only have 0–50 parasites/mL [69,70]. Symptomatic VL is characterized by a wasting illness with chronic fever, hepatosplenomegaly, and pancytopenia, and can result in death if not appropriately treated. Like malaria and American trypanosomiasis, symptomatic VL is a major global parasitic cause of morbidity and mortality [71]. Despite this, VL is an infection not commonly managed by American practitioners, a fact that could result in delayed recognition of or inappropriate therapy for undiagnosed reactivated VL, potentially leading to poor clinical outcomes.

Persons with asymptomatic VL are thought to greatly outnumber those with symptomatic VL by a factor of 4:1 in some locations (East Africa), and as high as 50:1 in others (Spain) [72,73]. We have established that potentially large numbers of previously deployed U.S. servicemembers [2] and likely immigrants from endemic global regions (see Table 2) with asymptomatic VL now reside in the U.S. Asymptomatic VL has medical relevance because of the potential for secondary transmission via blood transfusion and organ donation, the possible risk of domestic U.S. vector acquisition and subsequent transmission, and the reactivation risk associated with increasing use of immune modulating treatments and immunocompromising conditions. A review of *L. infantum* (syn. *chagasi*) worldwide presents varying rates of asymptomatic infection, with specific results depending on the geographic region and identifying assay used, but range from 5–54% [74,75,76]. Asymptomatic VL rates among blood donors in endemic areas of the Mediterranean are reported as between 1–22% [74].

The geographical distribution of VL in both the Old and New Worlds mirrors the endemicity of the implicated pathogens, specifically *L. donovani*, *L. infantum*, and *L. Mundinia* species. The greatest numbers of cases are reported from Brazil, India, Sudan, Ethiopia, and Kenya (Table 2) [77]. Globally, cases of VL have declined over the last decade; the World Health Organization’s (WHO) estimates between 50,000–90,000 VL cases annually, with a death rate of 95% in untreated cases [77]. The overall decline has been attributed primarily to VL control efforts made on the Indian subcontinent [78].

### 4.2. Asymptomatic VL: Immunity and Indicators of Progression

Asymptomatic visceral leishmaniasis (AVL) is variably defined, but it is considered present when a person demonstrates a positive *Leishmania* serological, culture, or nucleic acid-based test implying the presence of parasitic organisms in the absence of clinical signs or symptoms of active disease. Immune control rather than total eradication of the parasite is considered to be the most likely outcome following inoculation, so the presence of anti-leishmanial antibodies (including a positive rK39 assay or direct agglutination test [DAT]), positive interferon-gamma release assay (IGRA), leishmanin skin test, or polymerase chain reaction (PCR) are considered indicators of infection. Diagnosis of asymptomatic or subclinical infections can be obtained by histopathological methods and culture, though these more invasive tests are not recommended in the absence of clinically evident illness [73].

Accepting that asymptomatic persons with positive leishmaniasis tests have chronic infection and not just evidence of an immune response to prior infection implies that the host immune response required to achieve complete eradication is ineffective. *Leishmania* species achieve immune evasion and modulation by modifying cell signaling, surviving inside the phagosome, attenuating antigen presentation, and overall dampening of the normal immune response [83]. Indeed, immunomodulation is present from the first stages of infection, as demonstrated by studies showing the immunosuppressive effects of sand fly saliva and how its co-inoculation enhances the parasites’ ability to establish early infection [84]. The *Leishmania* parasite has been shown to persist in the spleen and bone marrow of mouse and non-human primate models [85,86,87]. A study of *L. infantum* infection in rhesus macaques showed that despite immunological control of early parasitemia, complete eradication from the reticuloendothelial system was not achieved, ultimately resulting in an inability to produce and sustain an effective, highly specific IgG antibody response, leading to parasite spread and disease progression [87]. Thus, in the presence of a reasonable exposure history and a positive leishmanial test result, it is prudent to assume that asymptomatic VL is present.

The natural history of VL may involve chronic asymptomatic infection (disease control), or progression to symptomatic disease. Studies in animal models have found that the interplay of IL-10, IL-12, INF-γ, and TNF-α was crucial to infection control [88]. Interleukin-10 signaling was needed for parasite persistence and latency, whereas IL-10 knockout mouse models are resistant to infection [87,89]. A Th1-type immune response driven by IL-12, INF-γ, and TNF-α is required for disease control; fatal leishmaniasis infection occurred in TNF-α knockout models [87,88].

Several studies assessed laboratory markers that correlated with progression from asymptomatic to symptomatic VL (Table 3). Among 1600 persons in endemic regions of India followed for three years by Chakravaty et al., 17 (1%) new cases of VL were identified. DAT, rK39, *Leishmania* IGRA, quantitative PCR, and genotyping were analyzed as possible biomarkers for progression to symptomatic VL. Those with a positive blood qPCR or strong positive DAT and/or rK39 assay results showed a statistically significant increased odds of progression to symptomatic disease (odds ratios of 20.9, 19.1, and 30.3, respectively); symptomatic VL tended to occur quickly after seroconversion (median 5 months) [90]. In another study, Das et al. determined immunological risk factors for AVL progression utilizing different parameters [91]. Screening 5794 persons from endemic villages in India with rK39, DAT, and qPCR blood testing, they determined the risk of progression to symptomatic disease based on how many of these individuals were positive for one, two, or all three of these markers. This study identified 42 persons with positive results on all three of these assays, and 23.8% of these individuals progressed to active VL over the course of 6 months (Table 3) [91].

In recent decades, the *L. infantum* outbreak in the southwest environs of Madrid, Spain illustrated important risk factors for VL, as well as the prevalence of asymptomatic infection. In the towns of Fuenlabrada, Leganés, Getafe, and Humanes de Madrid, the annual incidence of reported leishmaniasis infections rose >40-fold from 0.5 cases/100,000 persons per year in 2000–2009, to 22.2/100,000 persons per year between mid-2009–2012; the town of Fuenlabrada alone saw 43.5/100,000 cases per year [95,96]. 446 total cases of leishmaniasis were confirmed during the first three years, with VL comprising 35.9% (160 cases); this overall total subsequently increased to 758 by the beginning of 2018 [97]. Interestingly, persons who were identified as being of African origin developed a disproportionately greater amount of VL (89% of leishmaniasis cases among Africans were VL). 31.3% of VL cases were identified as having an underlying immunocompromised state; HIV was present in 10%, and 15.6% were taking some form of immunosuppressive medication [95].

An additional study on asymptomatic VL investigated 804 healthy persons from Fuenlabrada with no history of symptomatic leishmaniasis, using DAT, immunofluorescent antibody (IFAT), PCR, or a whole blood stimulation assay (WBA) with IL-2 quantification for detection of infection. Asymptomatic infection was defined as a positive result on any of these tests in the absence of signs or symptoms of active disease. The WBA-IL2 proved to be the most sensitive test, with 20.7% of the sample testing positive, as compared to 0%, 0.11%, and 1.0% of PCR, IFAT, and DAT tests, respectively [97]. This prevalence (20.7%) is similar to values noted for the leishmanin skin test (LST) when it was used as a screening tool for AVL in Georgia (19.3% LST positive) and Ethiopia (23.1%) [97,98,99].

### 4.3. Asymptomatic VL: Diagnostic Approach in the U.S.

Testing for asymptomatic visceral leishmaniasis is rarely performed outside of a research setting. Guidelines for the diagnosis of symptomatic leishmaniasis in North America have been published, though the specific assays that are favored may vary depending upon the specific clinical setting, disease endemicity, and available testing capabilities [100].

A practical approach in a low prevalence setting like the U.S. may involve using appropriate serologies for screening symptomatic persons coupled with direct parasite detection via histopathology, parasite culture, and/or PCR for confirmation. Of note, anti-leishmanial antibodies have been associated with cross-reactivity to other protozoan pathogens, including *T. cruzi* in humans as well as *T. gondii* (among others) in canines [101,102]. Serological assays may be less sensitive in those with immunocompromise (especially HIV) [103,104]. However, persons with AIDS often have higher burdens of parasitemia, increasing the sensitivity of parasitological diagnosis via microscopy and PCR testing [103,105,106].

The sole American population surveyed for asymptomatic VL was healthy U.S. servicemembers who had deployed to Iraq; the most frequently positive test (27/39; 69%) was the *L. infantum* interferon gamma release assay (IGRA) [2]. Unfortunately, the rK39 immunochromatographic test (approved by the Food and Drug Administration) has not been found helpful in asymptomatic *L. infantum* in studies in Brazil, Spain, and the U.S. [107,108].

There are currently three reference laboratories for leishmaniasis diagnostic testing in North America: McGill University in Montreal, Canada; the CDC in Atlanta, Georgia; and the Walter Reed Army Institute of Research in Silver Spring, Maryland (military beneficiaries only) [100]. Histopathology review, parasite culture, tissue PCR, rK39 serology, and species identification are offered. However, testing that would assist with diagnosis of asymptomatic VL (IGRA, WBA, blood PCR, Leishmanin skin test) is restricted in the U.S. to *Leishmania* IgG (an assay that has not been tested in any AVL surveillance) and perhaps the Karius test^®^.

Molecular testing (such as PCR amplification targeting leishmanial gene sequences) is of increasing importance [109]. Quantitative PCR can be performed on multiple tissue types including blood, and shows great promise with very high sensitivity and specificity (both >90%) [107]. This modality also appears to roughly correlate with parasite load, and with further study could potentially help discriminate between latent versus active subclinical disease [110]. Additionally, it may also be useful for monitoring response to therapy and identifying reactivation in immunosuppressed patients, due to the higher infectious burden and greater likelihood of circulating parasites in this population. Unfortunately, this test is not yet available in the aforementioned North American reference laboratories.

### 4.4. Reactivation, Prophylaxis, and Screening for AVL in Immunosuppressed Populations

There are several clinical settings where leishmaniasis poses significant risk for severe disease or reactivation. Historically, the most experience has been in advanced HIV disease, the risk for which has been greatly ameliorated with effective antiretroviral therapy and associated immune reconstitution. However, with increasing prevalence of organ transplantation and novel immunomodulatory therapies, other sources of immunosuppression are gaining importance. Reactivation is an emerging concern in the U.S. with a new pool of potentially thousands of AVL-infected veterans as well as immigrants from endemic regions (Middle East/North Africa [MENA], Latin America) who have ready access to and may one day require immunosuppressing therapies. Thus, the lifelong risk of reactivation, even remote from initial exposure, plays a critical role in this population.

#### 4.4.1. HIV-VL Co-Infection

The first incidence of HIV-leishmaniasis co-infection was reported in 1985, and a synergistic relationship has since been identified between the two infections. This is perhaps unsurprising, as infection and evasion of immune cells is central to both pathogens’ life cycle. VL increases HIV viral loads, may increase expression of latent virus, causes more rapid progression to AIDS, and reduces overall life expectancy [111,112,113]. HIV in turn increases the risk of both VL progression and mortality; indeed, whereas the mortality from VL in the setting of a transplanted organ has been reported to be around 22%, in a patient with poorly controlled HIV the mortality from VL rises to 25–46% [103,114,115]. HIV-VL co-infection increases the rate of progression to symptomatic VL by 100 to 2300-fold as compared to those without HIV infection, with the greatest risk occurring when CD4 T cell counts drop below 200 [111,115]. In southern Europe, between 5.6–16% of persons with HIV are suspected to have concomitant AVL [116].

#### 4.4.2. Solid Organ and Hematopoietic Transplantation

There are three ways in which a transplant recipient could acquire VL: first, due to transmission from undetected infection in the donor organ; second, via reactivation of a latent infection in the recipient once immune suppressed; and third, via de novo acquisition of a new infection in the post-transplant period. In the organ transplant population, it has been estimated that the overall risk of developing VL increases approximately 4-fold as compared to non-transplanted patients [117]. Data from the outbreak in Madrid have suggested that this figure may be even higher, with the risk of such an event in this cohort about 135 times greater than the immunocompetent population [118]. Incidence data for diagnosis of VL from 10 transplant centers in Spain and two in Brazil were acquired by retrospective record review of over 25,000 solid organ transplant patients between the years 1995–2012, and showed that 36 patients developed VL for an incidence of 0.1% [44]. A median of 11 months elapsed between transplantation and VL diagnosis (7 months in Brazil; 17 months in Spain), with kidney transplant recipients constituting the majority of affected persons (25 of 37 cases, 67.5%).

These findings are consistent with other data on transplant-associated VL, which tends to develop subacutely at a median of 6 to 19 months post-transplant depending on the organ involved (earlier for liver, later for kidney transplants) [114,117]. A review in 2018 by Akuffo et al. noted 148 total cases of transplant-associated VL, with 121 (82%) of those occurring in renal transplant recipients [118]. Overall prevalence of symptomatic VL in transplant recipients is estimated at around 0.05–0.9%, with the risk in non-VL endemic countries such as the U.S. probably lower [117,119]. Organ rejection and concomitant use of high-dose corticosteroids serve as prominent risk factors. Interestingly, transplant-associated VL may favor a more atypical presentation lacking some of the characteristic clinical features. Treatment and diagnosis are similar to persons without transplant, with liposomal amphotericin B being the favored therapy. Importantly, restoration of a functional immune response is a foundational part of treatment, and an effort should be made to reduce the intensity of immunosuppression in transplant patients with VL [100,119].

Data in the setting of hematopoietic transplantation is scarce. A case series reviewed 11 reported cases of VL, with a predominance of allogeneic over autologous transplant recipients (9 versus 2, respectively) [120]. VL was diagnosed by standard methods, and good treatment responses were achieved with liposomal amphotericin, although half of the patients with follow-up data available experienced relapse within 8 months. Interestingly, the onset of disease in this limited cohort occurred a median of 23 weeks following marrow transplant, faster than what has been reported for most solid organ transplant recipients [120]. The reason for this finding is unclear, however increased sampling (due to a low threshold for repeat bone marrow examinations in hematopoietic cell transplantation) may be a factor in shortening the time to diagnosis.

#### 4.4.3. Biologic Agents

Modern therapeutics have also seen the rise of novel immunosuppressive and immunomodulatory agents, many of which have been tied to VL activation [103]. Among these, tumor necrosis factor-α (TNF-α) antagonists bear special mention. There are currently five such agents in use in the U.S. (etanercept, infliximab, adalimumab, certolizumab pegol, and golimumab), and their mechanism of action provides for a plausible immunologic means for interaction with leishmanial infection. TNF-α is a critical cytokine for the development of a strong Th1-mediated immune response, and it is primarily produced by macrophages to stimulate multiple other cell types including phagocytes and T lymphocytes; it also plays a key role in granuloma formation.

Visceral leishmaniasis occurring in the setting of TNF-α administration has been recognized in case reports from Europe since at least 2004 [121,122]. One review of such patients recounted 32 cases of leishmaniasis (16 visceral) occurring in or being directly traced to endemic regions [121,123,124,125]. Additional epidemiology comes from studies of at-risk populations in Italy and Spain looking at the prevalence of positive leishmaniasis testing in patients on TNF-α inhibitor therapy. Amongst a cohort of 50 patients on biologic agents in northern Italy (compared to matched cohorts of immunocompetent persons and persons on non-biologic therapies) there were 18 patients (36%) with PCR evidence of underlying *L. infantum* infection; three of them had parasite burdens of greater than 1,000,000 parasites/mL, yet were still asymptomatic This was a statistically significant difference as compared to the immunocompetent group, where only 4 of 50 patients (8%) were PCR positive. Interestingly, no patient on non-biologic therapies tested positive for leishmanial infection [126]. Another study examined the prevalence of asymptomatic *L. infantum* infection in a cohort of 192 inflammatory bowel disease patients taking TNF-α inhibitors in Catalonia, Spain. This study defined AVL as positive anti-leishmanial antibodies and/or peripheral blood PCR, and found that the overall prevalence of asymptomatic disease was 10.9% (3.1% antibody positivity; 8.8% PCR positivity) [127].

Several common themes appear to have emerged, including an association with all forms of leishmaniasis, a generally typical disease presentation, a prolonged interval period between TNF-α inhibitor initiation and symptomatic disease onset (median of 18 months in one series), and a relatively greater risk associated with monoclonal antibody constructs (e.g., adalimumab, infliximab) as compared with the pseudo-TNF receptor etanercept [128,129]. This latter phenomenon may in part be related to the ability of monoclonal antibodies to inhibit both soluble and transmembrane forms of TNF-α, whereas etanercept can only neutralize the soluble molecule, resulting in somewhat less potency and a concomitant lower risk of reactivation and dissemination [130].

Treatment guidance for TNF-blockade associated VL is limited to the experience cited in case series; good outcomes with relatively low recurrence rates have been attained with standard anti-leishmanial therapies coupled with reduction in or temporary cessation of immunosuppression, and over 90% of patients can achieve clinical cure with this approach [125,129]. Zanger et al. reported leishmanial recurrence in two of seven rechallenged patients (one cutaneous and the other mucosal leishmaniasis), with no relapses of VL in the other five. Among 49 TNF-alpha inhibitor recipients with cutaneous, mucosal, and visceral leishmaniasis (predominately cutaneous) in the Mediterranean region; relapse was uncommon (5 out of 49 patients), but it was associated with a failure to discontinue the patient’s immunosuppressive agent [119]. Whether anti-TNF agents can safely be reinstituted following treatment of an initial episode of VL remains unclear at this time, and should be done with close clinical monitoring.

### 4.5. VL Prophylaxis and Screening within the U.S.

#### 4.5.1. VL Primary and Secondary Prophylaxis

There are no recommendations for primary prophylaxis of VL within the U.S. at this time [131,132,133,134,135]. U.S. guidelines specifically recommend against prophylactic or pre-emptive therapy [118,133,134,135]. While recognizing the threat that reactivation of VL may pose, there is presently no data to support clear benefit from implementation of a prophylactic regimen in the absence of prior active disease. Factors that contribute to this stance in the U.S. include the extremely low historical prevalence and the toxicity of available treatments. Reactivation risk in persons with AIDS is drastically reduced with the initiation of effective antiretroviral therapy (ART), thus U.S. guidelines endorse starting ART as soon as possible, even when underlying VL is present or suspected. Indeed, although an immune-reconstitution syndrome has been described in case reports of HIV-VL co-infection, there are no reports suggesting worse outcomes following prompt antiretroviral initiation [136].

There is, however, a clear role for secondary prophylaxis of VL in HIV patients. In persons with HIV not receiving ART who have a prior episode of VL, the relapse rate at one year is as high as 90%; however, with effective prophylaxis this can be significantly reduced [116,137]. Use of either amphotericin or pentavalent antimony are options, with a liposomal formulation of amphotericin given once every 2–4 weeks being preferred in the U.S. Secondary prophylaxis can likely be stopped after immune reconstitution (CD4 count >200) and virologic suppression [133,138]. Though the relapse rate in solid organ transplant is estimated to be as high as 25%, secondary prophylaxis is not recommended in persons who have undergone solid organ transplantation until a first relapse has occurred [139]. Secondary prophylaxis in the setting of TNF-α inhibitors has not been described. Persons with a history of VL during immunosuppression require prolonged (and likely lifelong) monitoring, as neither treatment nor secondary prophylaxis can fully prevent the possibility of further relapses [100].

#### 4.5.2. Screening

Testing for asymptomatic leishmanial infection is not currently recommended in any U.S guideline prior to organ transplantation, during the evaluation of newly diagnosed persons with HIV, or before initiation of immunosuppressive medications [132,133,134,135]. This is largely due to lack of available data to support the benefit of such a practice. Additionally, because the exact role that asymptomatic persons may play in ongoing chains of transmission remains to be elucidated (see Section 5.3), the benefit of screening within this population is yet to be determined. Further studies clarifying the role of AVL within a population are needed, and could potentially influence the current management practices for persons living with this entity. Currently, known VL seropositive status in an organ donor or recipient is not a contraindication to transplantation in the U.S.; close clinical monitoring for reactivation following transplantation is advised [100,132].

## 5. Prevention, Secondary Transmission, and U.S. Vector-Borne Transmission of VL

As there is no VL vaccine available, prevention efforts focus on minimizing the risk of vector exposure while traveling in endemic regions by limiting outdoor activities at night when sand flies are most active, wearing long-sleeved clothing, and the use of bed nets and insect repellent [140].

### 5.1. Blood Product Transfusions and Visceral Leishmaniasis

Blood products are not screened for leishmaniasis in the U.S., however transmission of leishmaniasis has been documented via blood transfusion [141,142]. Two reviews discussed 13 reports, with most instances occurring in children age six or younger; both *L. donovani* and *L. infantum* were implicated [141,143]. Serosurveillance of Brazilian dialysis patients found a VL prevalence of 5.4–9% overall, but as high as 25% in some areas [144,145]. Among multiply transfused hemodialysis patients in one series, the seroprevalence reached 37%, and after follow-up of 27 seropositive patients, four developed symptoms of active VL [144]. Within France, 41.4% of asymptomatic blood donors screened positive for VL in a recent single-center study [146]. Such statistics pose a quandary within the U.S., particularly with regard to the military practice of using a “walking blood bank” in the deployed environment to quickly administer massive whole blood transfusions in the setting of severe war trauma. Though such actions can be life-saving in the acute period, the risk of potential transfer of cell-associated pathogens like VL species during resuscitation in endemic regions has not been assessed.

Methods for mitigating the risk posed by leishmanial parasite contamination of blood have been proposed [143]. One strategy relies upon leishmanial parasites residing within macrophages, therefore making them susceptible to removal by leukoreduction filters. Use of leukoreduction filters on blood samples which were positive for leishmanial DNA was able to revert these samples to PCR-negative status following filtration [147]. Other techniques that have shown promise include riboflavin or the psoralen-related compound amotosalen coupled with UV light treatment, as well as thiopyrylium treatment [143].

### 5.2. Transplant-Related Transmission

Leishmaniasis transmission could occur via pre-existing asymptomatic infection in an organ donor, which is shared with the recipient during transplantation. The risk of an infection-naïve patient acquiring VL from an infected donor organ is not known, mainly due to a lack of screening of both organ donors and recipients for asymptomatic leishmanial infection. A case report from the United Kingdom illustrated this principle; after a liver transplant patient in the U.K. developed worsening cytopenias, a bone marrow biopsy revealed the presence of *Leishmania* amastigotes. As the patient had no apparent history of exposure to endemic regions, the donor’s serum was tested and returned positive for leishmaniasis (donor was noted to have a history of travel to India) [148]. Review of all reported cases of parasitic disease occurring in post-solid organ transplant patients from 1996–2016 revealed 151 cases of leishmaniasis; one case (0.7%) was determined to represent post-transplant de novo acquisition, five cases (3.3%) were considered reactivation of latent infection in the recipient, and the origin of the remainder (96%) was unknown [149]. Reassuringly, although the risk of VL acquired via organ transplant is yet to be fully defined, available data supports that transplant-related VL is uncommon, even in leishmaniasis endemic regions [139].

### 5.3. AVL and Potential Domestic Transmission via U.S. Sand Flies

*Lu. shannoni* is a competent sand fly vector of *Leishmania infantum* and is widely distributed in the U.S. [26]. This raises the spectre that introduction of *L. infantum* (possibly by asymptomatic, previously deployed U.S. military servicemembers, or infected hunting and companion dogs) could lead to vector borne transmission of VL within the U.S. To better understand the risk that persons with AVL may pose with regard to anthroponotic spread, several studies have been performed within endemic countries using xenodiagnosis. Molina et al. in the wake of the aforementioned Madrid outbreak identified four groups of persons with current or previous VL: asymptomatic immunocompetent persons (14 with AVL, 10 with no evidence of prior VL), untreated persons with symptomatic VL, formerly symptomatic VL patients post-treatment, and three immunocompromised patients with VL (two persons with HIV, one on therapy for multiple myeloma) [150]. None of the 14 asymptomatic VL individuals were found on xenodiagnosis to transmit infection to the sand flies. Amongst persons with a history of VL, none of those who had received treatment permitted VL transmission to sand flies, however two individuals with active but untreated disease were able to transmit; notably, most of this transmission occurred in one individual with a high parasite load of 1050 parasites/µL as measured by quantitative PCR. The most effective reservoirs for transmission were the immunosuppressed persons. The risk of transmission correlated with parasite burden, with transmission frequency reaching as high as 30.6% in the patient with hematologic malignancy whose parasite load was 4360/µL prior to VL treatment, and 48.6% in one of the two HIV/VL co-infected individuals (608 parasites/µL). Strikingly, this latter individual continued to have detectable parasitemia and showed anthroponotic transmission potential even while taking anti-retroviral therapy and following treatment and institution of secondary prophylaxis for his *Leishmania* infection [150]. Previous work in six HIV-VL infected persons showed an association between low CD4 count and infectivity to sand flies [151,152]. Further weight to this finding is provided by Ferreira et al. in Brazil, who examined 61 persons for their ability to transmit *L. infantum* to the primary local vector, *Lu. longipalpis*. The studied cohort was divided into four groups: AVL with and without HIV co-infection (2 and 18 persons, respectively), and symptomatic VL with and without HIV co-infection (20 and 20 persons). Positive xenodiagnosis of sand fly infection was confirmed by positivity on microscopy and/or PCR. Amongst the persons without HIV, those with symptomatic VL were significantly more infective to sand flies that those who were asymptomatic. Amongst symptomatic patients, those with HIV were more likely to be positive by microscopy, however PCR-positivity was equal between both HIV-infected and HIV-uninfected groups. Notably, however, the HIV infections in this population were relatively well-controlled. Although the final analysis concluded that both symptomatic disease and HIV infection were associated with an increased ability to transmit infection to sand flies, perhaps the most interesting finding was that 5.4% of sand flies fed on asymptomatic, HIV-uninfected persons were found *Leishmania* PCR positive. Though sand fly infection could not be confirmed by microscopy, this finding suggests that humans with AVL and no underlying immunosuppression may be able to act as reservoirs, with the potential for low levels of anthroponotic spread [150].

Taken together, these results demonstrate the potential for anthroponotic transmission, with the greatest risk stemming from parasitemic immunocompromised persons [151]. However, although anthroponotic spread is possible, more research is needed before a realistic determination as to the risk of this occurring within U.S. borders is made. Specifically, xenodiagnostic studies using *Lu. shannoni* would be particularly illuminating.

## 6. Possible Long-Term Risks of Asymptomatic Visceral Leishmaniasis?

Uncertainty surrounds our knowledge of what effects subclinical, chronic VL might have years and even decades after initial infection. Chronic inflammation, regardless of etiology, often has deleterious effects on health. The chronic inflammatory state is broadly implicated in the dysfunction of multiple organ systems and has been identified as a strong risk factor for the development of many common causes of morbidity and mortality, including cardiovascular disease, diabetes, malignancy, chronic kidney disease, and neurodegenerative disorders such as Alzheimer’s dementia [153]. Indeed, aging appears to be accompanied by a progression towards upregulation of certain inflammatory pathways, which may contribute significantly to disease in older populations [154]. Thus, it can be implied that the additional inflammatory burden imposed by chronic infection may hasten these processes, and while there are many causes of chronic inflammation, the unique relationship that leishmaniasis shares with the immune system perhaps constitutes a particular risk for long-term consequences.

Fundamentally, human leishmaniasis is an infection of macrophages. Macrophages, in turn, are one of the most active components of our immune system, responsible not only for phagocytosis of pathogens but also recruitment and activation of other inflammatory cells critical to both the innate and adaptive immune responses. Animal models have demonstrated features of the aberrant immune response engendered by visceral leishmaniasis, however longitudinal studies examining the more remote macroscopic consequences of this immune escape and dysregulation are lacking [155].

Though the effects that chronic asymptomatic visceral leishmaniasis may have on humans remain to be defined, there are examples of other chronic immunomodulatory illnesses that are known to produce either poor or co-morbid outcomes. With effective ART, HIV has become a prototypical example. Indeed, even when HIV viral replication is controlled, other non-AIDS-related illnesses (including cardiovascular disease, neurocognitive disease, osteoporosis, liver disease, kidney disease, and non-AIDS defining cancers) occur at an increased rate in this population [156,157,158]. As our understanding and ability to treat this disease has advanced, so too has the need to recognize, address, and effectively manage other important long-term issues [156,159,160].

The same is likely true of leishmaniasis. Like HIV, leishmanial infection involves chronic infection of and residence within the immune system, and parasite persistence necessarily requires immunomodulation characterized by deviation away from appropriate immunologic responsiveness toward a pattern involving both abnormal stimulation and immune tolerance [161,162]. Could we still be yet to see the impact of this chronic inflammation among thousands of U.S. veterans, with unrecognized asymptomatic VL someday leading to “premature aging” consequences? While the ultimate outcome and clinical significance of this are yet to be elucidated, it is a question that deserves further study as progress is made in the treatment and control of visceral leishmaniasis.

## 7. Conclusions

Though great strides have been made in some parts of the world, leishmaniasis continues to represent a significant global cause of morbidity and mortality that disproportionately affects the underserved and impoverished. We have been fortunate here in the U.S. that this disease has remained uncommon, with endemically acquired human cases being restricted thus far to cutaneous forms of *L. mexicana*. However, a number of factors provide cause for concern and increased vigilance regarding the future potential for greater incidence of both cutaneous and visceral leishmaniasis in the previously low prevalence setting of the United States. Climate change, the presence of competent reservoirs and vectors, significant population subsets with clear exposure/infection risks (Latin American immigrants and previously deployed military servicemembers), and widespread use of advanced medical therapies such as organ transplant and potent immunosuppressive medications all combine to create an environment where increased acquisition of domestically encountered leishmanial infections is a plausible possibility. The first (and perhaps most fundamental) step in combating such an event is raising awareness of this disease amongst a population of healthcare providers with little experience in its recognition, diagnosis, and treatment. We provide such a summary, with the goal that patients and physicians both within the U.S. and abroad may benefit from this review of recent developments, emerging challenges, and new thinking in this field.

## Figures and Tables

**Figure 1 microorganisms-09-00578-f001:**
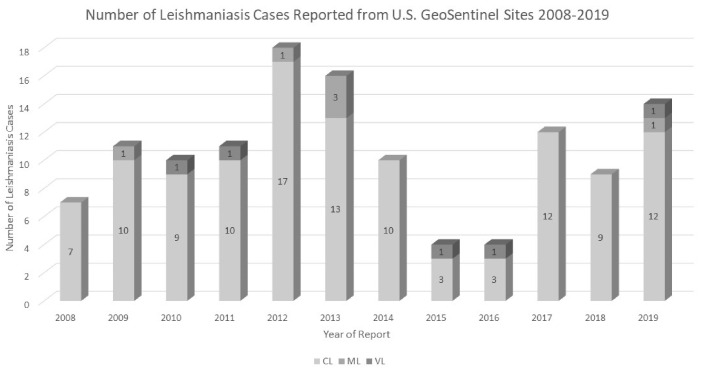
Number of Returned Travelers Reported from a U.S. GeoSentinel Surveillance Network Site with a Diagnosis of Leishmaniasis from 2008–2019. CL = cutaneous leishmaniasis, ML= mucosal leishmaniasis, and VL = visceral leishmaniasis. (Data kindly provided courtesy of Dr. Michael Libman and the GeoSentinel Surveillance Network)**.**

**Table 1 microorganisms-09-00578-t001:** U.S. Sand Flies that may Transmit Leishmaniasis.

Sand Fly Species	U.S. Geographic Locations	Competent *Leishmania* Vector
*Lu. anthophora*	Arizona, Oklahoma, Texas	*L. mexicana* [22]
*Lu. cruciata*	Florida, Georgia	*L. mexicana* [28]
*Lu. diabolica*	Texas	*L. mexicana* [27]
*Lu. shannoni*	Widespread; AL, AR, DE, FL, GA, KY, KS, LA, MD MO, MS, NC, NJ, OH SC, TN, TX [26]	*L. mexicana* [27] *L. infantum* [18,20] ? *L. major* [25]—transmission not proven, can be infected

**Table 2 microorganisms-09-00578-t002:** *Leishmania* Species Causing VL and Their Geographic Distribution * ± [77,79,80,81].

*Leishmania donovani* Includes species formerly known as *L. archibaldi.*	East Africa and Southern Arabia **Sudan, Ethiopia, Eritrea, Kenya,** Uganda, **Somalia, South Sudan**
	Northwestern China Xinjiang Autonomous Region
	South Asia **India**, **Bangladesh**, Nepal, Sri Lanka, Pakistan
*Leishmania infantum* (synonym: *L. chagasi*)	Central and South America Primarily **Brazil**; also Argentina, Paraguay, Colombia, Venezuela, Honduras, Guatemala, Bolivia, Mexico, Uruguay
	Arabian Peninsula Yemen and Saudi Arabia
	Mediterranean, North Africa, and Middle East Spain, France, Greece, Italy, Portugal Tunisia, Balkans, Algeria, Libya Israel, Turkey, Iran, Iraq, Kuwait, Syria
	Western Asia and China Afghanistan Provinces of Gansu, Shaanxi, Shanxi and Sichuan; Xinjiang Region
*L. (Mundinia)* species Primarily *L. martiniquensis*; additional genus members include *L. enriettii* complex, *L. orientalis*	Thailand, Myanmar, Grenada, Martinique

* List is not all encompassing; per WHO, VL is endemic in ~79 countries worldwide [82]. ± Bolded nations estimated to harbor >90% of the global burden of VL.

**Table 3 microorganisms-09-00578-t003:** AVL and Biomarkers for Progression to Symptomatic VL.

Study Name (Year) [Reference]	Location	Species	Study Size	Tests Used	Follow-Up Duration	Risk of Progression	Factors Associated with Risk of Progression
Evans et al. (1995) [92]	Brazil	*L. infantum*	653 (children)	Anti-leishmanial antibodies	5 years	6.1%	Seroconversion; living in household with prior VL case
Hasker et al. (2014) [93]	India, Nepal	*L. donovani*	32,529	rK39, DAT	1 year	6.4% (India; high baseline DAT)	High titers of rK39 and/or DAT; new seroconverters
						9.8% (Nepal; high baseline DAT)	
						7.3% (India; high baseline rK39)	
						7.7% (Nepal; high baseline rK39)	
						12.5% (India; new seroconversion)	
						9.1% (Nepal; new seroconversion)	
Chapman et al. (2015) [94]	Bangladesh	*L. donovani*	2410	rK39	3 years	14.7%	High titer rK39, especially in new seroconverters
Chakravaty et al. (2019) [90]	India	*L. donovani*	1606	rK39, DAT, IGRA, qPCR	3 years	1.6% (8/476 known seroconverters)	High titer DAT; high titer rK39; +qPCR
Das et al. (2020) [91]	India	*L. donovani*	5794	rK39, DAT, qPCR	6 months	3.27% (+ rK39 only)	+ for mid-high titer rK39 and DAT plus + qPCR
						8.33% (+rK39 and DAT)	
						23.8% (+rK39, DAT, and qPCR)	

## Data Availability

Not applicable.

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
