# Peer review of "Leishmaniasis in the United States: Emerging Issues in a Region of Low Endemicity"

_microorganisms, 2021, doi:10.3390/microorganisms9030578_

Round 1

Reviewer 1 Report

The authors address the point of Leishmaniasis in the US in the past, the present and give a relatively good perception of the highly probable threat in the future. They address the causes of that threat, climate change making more states favorable for the installation of competent vector species associated with an entry of asymptomatic carrier through the military coming back from endemic countries.

Since climate change is unlikely to be overturned in the actual world politics, this review is timely.

Though I personally find it hard to read, all necessary information is here to give an overview of the situation and good perspectives.

Comments:

Lime 77-82: I would argue on the species identification of the North Dakota case. Nothing in the case report shows that L. infantum is involved. L. infantum is not present in Nepal, the endemic species is L. donovani. L. donovani can give CL but so far, those strains have been only identified in Sri-Lanka and seem to have a mutation in virulence factors. The North Dakota case would require WGS in order to clarify the situation, but we can not clearly identify L. infantum with the actual data.

A more general comment, on prevention. The authors clearly say that prophylaxis is clearly not recommended in absence of a symptomatic infection. This is particularly due to the fact that we don’t know if asymptomatics are part of active transmission chains because the research in that matter is costly and complex when it comes to leishmaniasis. A comment on this particular aspect would be appreciated. Answering that question ASAP could change the PH guidelines for asymptomatic carrier treatment therefore further helping countries that are controlling the disease (before COVID) and help planning a strategy for the future handling of the disease complex in the North.

Author Response

Responses to Reviewer 1:

Comment 1 -

Line 77-82: I would argue on the species identification of the North Dakota case. Nothing in the case report shows that L. infantum is involved. L. infantum is not present in Nepal, the endemic species is L. donovani. L. donovani can give CL but so far, those strains have been only identified in Sri-Lanka and seem to have a mutation in virulence factors. The North Dakota case would require WGS in order to clarify the situation, but we can not clearly identify L. infantum with the actual data

Response: The authors thank the reviewer for noting our error and agree that L. donovani is the likely species. As the PCR done at CDC cannot discern L. donovani from L. infantum, we think it is most precise to modify to L. donovani-infantum in this sentence.  While it is not common for CL in India, Nepal, or Bhutan to have species identification performed there are reports outside of Sri Lanka, as well as our own clinical experience, that localized CL can be caused by L. donovani. (references such as Thakur L, EID Aug 2020, Thakur L, Frontiers Cellular and Infection Micro 2020, Pal A Ind J Derm Ven Leprol 2020).

Revision: In line 79, L. infantum was changed to L. donovani-infantum to reflect this.

Comment 2 -

A more general comment, on prevention. The authors clearly say that prophylaxis is clearly not recommended in absence of a symptomatic infection. This is particularly due to the fact that we don’t know if asymptomatics are part of active transmission chains because the research in that matter is costly and complex when it comes to leishmaniasis. A comment on this particular aspect would be appreciated. Answering that question ASAP could change the PH guidelines for asymptomatic carrier treatment therefore further helping countries that are controlling the disease (before COVID) and help planning a strategy for the future handling of the disease complex in the North.

Response: The authors thank the reviewer for this important perspective. Though we review data regarding our current knowledge of asymptomatic transmission in section 5.3, we previously did not tie this directly to our discussion in section 4.5. We have added several lines making this connection to better tie these sections, and the clinical relevance, together.

Revision: The following was added from lines 548 – 553: “Additionally, because the exact role that asymptomatic persons may play in ongoing chains of transmission remains to be elucidated (see section 5.3), the benefit of screening within this population is yet to be determined. Further studies clarifying the role of AVL within a population are needed, and could potentially influence the current management practices for persons living with this entity.”

Reviewer 2 Report

The review is well written, extensive in its coverage with 160 references and is important and valuable to public health medical workers and clinical doctors in the US.

What follows are only suggestions that might add to this extensive review.

312 When describing the many ways Leishmania evade the immune response, it would be pertinent to also mention that the sand fly injects its saliva which is greatly immunosuppressive.

242 When discussing treatment, FDA approved radiofrequency-induced heat therapy  for CL might be included with a reference by the senior author:

Aronson N. E. et al. 2010 a randomized control trial of local heat therapy versus intravenous sodium stibogluconate for the treatment of cutaneous Leishmaniasis infection. PLoS  Neglected Tropical Diseases 4(3) e628

David J.R. 2018 The successful use of radiofrequency-induced heat therapy for cutaneous leishmaniasis: a review. Parasitology 145, 527-536

417 When discussing HIV-VL co-infection, might refer to  the paper that describes several patients with CL and HIV whose CL did not respond to multiple treatments with sodium stibogluconate but did to radio-frequency-induced heat treatment.

Prasad Net al. 2011. Heat, oriental sore and HIV. Lancet 377, 610

Author Response

Responses to Reviewer 2:

Comment 1 –

312 When describing the many ways Leishmania evade the immune response, it would be pertinent to also mention that the sand fly injects its saliva which is greatly immunosuppressive.

Response: The authors thank the reviewer for this comment, and have added a sentence mentioning this phenomenon to the relevant section (section 4.2) with an appropriate reference (reference 83).

Revision: Lines 315 – 318 were added as follows: “Indeed, immunomodulation is present from the first stages of infection, as demon-strated by studies showing the immunosuppressive effects of sand fly saliva and how its co-inoculation enhances the parasites’ ability to establish early infection [83].”

Comment 2 -

242 When discussing treatment, FDA approved radiofrequency-induced heat therapy for CL might be included with a reference by the senior author.

The authors have added mention of thermotherapy as referenced by Dr. David’s recent review to their brief section on new developments for leishmaniasis treatment. The addition of the second reference mentioned by the reviewer was not included, as we felt that the use of Dr. David’s paper alone should be sufficient. This reference was added as reference number 61.

Revision: Lines 248-249 now read, “Photodynamic therapy and thermotherapy continue to look promising for treatment of Old World cutaneous leishmaniasis.” Reference 61 was added.

Comment 3 -

417 When discussing HIV-VL co-infection, might refer to the paper that describes several patients with CL and HIV whose CL did not respond to multiple treatments with sodium stibogluconate but did to radio-frequency-induced heat treatment.  Prasad Net al. 2011. Heat, oriental sore and HIV. Lancet 377, 610”

The authors thank the reviewer for this comment, however the section here pertains to issues surrounding visceral leishmaniasis reactivation, prophylaxis, and screening (section 4.4), as opposed to issues surrounding treatment of patient with HIV-CL co-infection, thus we would prefer not to reference this important paper on HIV-CL treatment.